# Fabrication and Evaluation of Tubule-on-a-Chip with RPTEC/HUVEC Co-Culture Using Injection-Molded Polycarbonate Chips

**DOI:** 10.3390/mi13111932

**Published:** 2022-11-09

**Authors:** Ju-Bi Lee, Hyoungseob Kim, Sol Kim, Gun Yong Sung

**Affiliations:** 1Interdisciplinary Program of Nano-Medical Device Engineering, Hallym University, Chuncheon 24252, Korea; 2Integrative Materials Research Institute, Hallym University, Chuncheon 24252, Korea; 3Major in Materials Science and Engineering, School of Future Convergence, Hallym University, Chuncheon 24252, Korea

**Keywords:** tubule-on-a-chip, metformin, RPTEC, HUVEC, microfluidic chip

## Abstract

To simulate the ADME process such as absorption, distribution, metabolism, and excretion in the human body after drug administration and to confirm the applicability of the mass production process, a microfluidic chip injection molded with polycarbonate (injection-molded chip (I-M chip)) was fabricated. Polycarbonate materials were selected to minimize drug absorption. As a first step to evaluate the I-M chip, RPTEC (Human Renal Proximal Tubule Epithelial Cells) and HUVEC (Human Umbilical Vein Endothelial Cells) were co-cultured, and live and dead staining, TEER (trans-epithelial electrical resistance), glucose reabsorption, and permeability were compared using different membrane pore sizes of 0.4 μm and 3 μm. Drug excretion was confirmed through a pharmacokinetic test with metformin and cimetidine, and the gene expression of drug transporters was confirmed. As a result, it was confirmed that the cell viability was higher in the 3 μm pore size than in the 0.4 μm, the cell culture performed better, and the drug secretion was enhanced when the pore size was large. The injection-molded polycarbonate microfluidic chip is anticipated to be commercially viable for drug screening devices, particularly ADME tests.

## 1. Introduction

Kidneys exist in pairs on the dorsal side of the stomach, excreting wastes and maintaining homeostasis in the body, and they have three main functions. First, there is the excretory function of filtering metabolites and wastes and excreting them in the urine, second, the maintenance of body homeostasis keeps the amount of water, electrolytes, and acidity in the body constant within a narrow range, and the most important feature of the kidney function is the filtering and reprocessing of substances. absorption and excretion. As the filtrate passing through the glomerulus and Bowman’s sac passes through the proximal tubule, about 80% of most nutrients such as glucose and amino acids and sodium ions (Na^+^), potassium ions (K^+^), chlorine ions (Cl^−^), carbonate ions (HCO^−^), and other essential electrolytes are reabsorbed, and waste products present in capillaries without passing through Bowman’s sac are secreted back into capillaries. Protein, albumin, and blood are rarely found in the urine of healthy people because the components our body needs are reabsorbed in the proximal tubule. However, if there is a problem with the reabsorption function, protein, albumin, blood, etc. may be seen in the urine, and waste materials accumulate in the body, causing uremia and kidney disease (nephropathy).

Recently, by developing a microphysiological system using an organ-on-a-chip, efforts are being made to solve the problems of the existing drug development method, which requires huge development costs for a long time and the limitations of animal experiments due to differences between species [1,2,3]. Organs under study are being studied in various ways, such as Lung-on-a-chip [4], Glomerulus-on-a-chip [5], and Heart-on-a-chip [6].

Chip research on proximal tubules is also being actively conducted. In various studies, patterns are made using soft lithography, and research using tubule-on-a-chip made with PDMS is in progress [7,8]. These studies attempted to reproduce the proximal tubular structure of the renal nephron by culturing RPTEC in a single perfusion chamber with the aim of mimicking the function of renal tissue [9,10]. In another study, tubule-on-a-chip was fabricated by using 3D bioprinting to form a gasket using silicone elastomer by placing gelatin and fibrin as ECM to form the proximal tubule structure in the chip [11]. In addition, gene expression analysis, cytokine secretion, permeability analysis, and the expression of cyclosporin A were observed using RPTEC and RPTEC/TERT1 cells, and the structure of the proximal tubule was investigated.

To reproduce organs using chips, calculating the chip shear stress and creating conditions similar to the actual organ shear stress is necessary. Many studies have also been conducted on the effects of seamless kidney chips [12,13]. Due to the movement of urine, tubular epithelial cells in the kidney are subjected to fluid shear stress and are continuously exposed to the fluid. Therefore, it is important to implement the fluid shear stress (FSS) in the test tube and provide an FSS that does not cause kidney damage. For in vitro studies conducted using FSS, an environment similar to the proximal tubule of the kidney and appropriate culture conditions were selected and shear stress was given [14].

In many studies, PDMS is used to produce microfluidic chips. Although PDMS has advantages when fabricating chips with high transparency, biocompatibility, low fluorescence, chemical inertness and high gas permeability, it has the disadvantage that small hydrophobic molecules diffuse and absorb very quickly into the PDMS. This limits their use in some biological applications, including cell culture experiments that require drugs with intracellular targets [15]. In order for a drug to penetrate the cell membrane and reach the inside of the cell, the molecule must be small (MW < ≈500 Da) and hydrophobic, because this property is easily adsorbed by PDMS. Therefore, it is difficult to accurately predict the concentration of the drug inside the microfluidic channel as the drug is absorbed and lost to the PDMS.

To address the issue of drug absorption, which is a disadvantage of PDMS, a polycarbonate (PC) chip was used in this experiment. PC is a durable material formed by the polymerization of bisphenol A and phosgene, which results in the formation of repeated carbonate groups. PC is suitable for use in DNA heat circulation owing to its transparency in visible light and extremely high glass transition temperature (≈145 °C) [16]. Other advantages of PCs include their low cost, high impact resistance, low moisture absorption, and excellent processing characteristics [17]. In addition, unlike PDMS, the mass production of PC is feasible, which is expected to be a significant advantage for future commercialization. Three-dimensional (3D) chips have a greater physiological relevance than 2D cell cultures, and they can employ ethical methods that do not involve animal testing in vitro [18]. Furthermore, as it will be possible to mass-produce chips using injection-type PC chips in the future, we believe that their commercialization will facilitate and simplify research. In this study, the ADME process including absorption, distribution, metabolism, and excretion in the human body was simulated after drug administration, and a kidney reabsorption–secretion model was constructed and evaluated using an injection-mold microfluidic chip.

## 2. Materials and Methods

### 2.1. Cell Culture

Human RPTEC (Renal Primary Tubule Epithelial Cell) (LONZA, CC-2553) was cultured through an REGM BulletKit (LONZA, SWISS, Basal) and used within passages 5–7. HUVEC (Human Umbilical Vein Endothelial Cell) (LONZA, C2519A) were cultured through EGM-2 KIT (LONZA, SWISS, Basal) and used within passage numbers 4–7. All cells were cultured from a 75-Tflask to approximately 90% of the total area, which was reproduced at a concentration of 3 × 10^6^ cells/mL, and seeded into membranes. The badges were replaced once every 1–2 days.

### 2.2. Fabrication of Injection-Molded Chip (I-M Chip)

Figure 1 shows the schematic of the I-M chip (K-bio, Osong, Korea) that consists of a single layer of PC material with three badge chambers and separate-fit inserts at each end of a single channel. In addition, a single type of tissue cell is cultured simultaneously with three inserts, and three samples are obtained from one chip to aid with statistical results processing. PC is selected as the material for the chip due to its superior processing characteristics and low adsorption properties [17].

After an injection-molded PC layer is adhered to a glass using double-sided tapes (3M, Saint Paul, MN, USA), so as not to cause bubbles, an O-ring is sandwiched in a part where an insert is fitted. For the insert, EB-107LP-2 Part A and B (EpoxySet, Woonsocket, RI, USA) are mixed at a 100:35 ratio, and an epoxy adhesive is applied thinly to the insert before it hardens. This is followed by the application of membranes with a pore diameter of 0.4 μm or 3 μm (It4ip, Belgium, Louvain). After hardening the epoxy for one day so that the insert and membrane adhere, the epoxy was sterilized in an autoclave at 121 °C for 5 min.

In a culture method for pasting a PC chip onto a glass and culturing it with an insert, RPTEC was cultured on the upper side of HUVEC under the membrane standard. The flow was directed to the HUVEC side; then, the blood flow of blood vessels was described, and secretion from HUVEC to the RPTEC side was evaluated. FSS is applied to the HUVEC through the channel to assist in the biological representation of a blood vessel. Consequently, the function of incubating the RPTEC on the membrane and secreting it from the HUVEC to the RPTEC can be evaluated. ECM is a non-cellular three-dimensional structure formed by collagen type 1.

Utilizing the fact that the chip is not an integrated structure but rather a separable insert-type structure, the insert is placed oppositely so that the membrane can go upward before mounting the insert on the chip. HUVEC is first seeded on the lower side of the membrane for a day. After the HUVEC layer is formed, the insert is attached to the chip so that the HUVEC is below the membrane, and ECM is placed on the upper side of the membrane before RPTEC is cultured. A suitable mixture of REGM and EGM2 was used for empty culture. In order to give the chip the same shear stress as the proximal tubule in vivo, it should be cultured by giving flow by giving it an angle. When modeling a tubule on a chip, in other studies, it is usually cultured by applying a shear stress of 0.1 to 5 dyne/cm^2^ [19,20]. The normal shear stress range in the rat model was validated as 0.5 to 5 dyne/cm^2^ [21]. Cell growth was better when the shear stress applied to the cells was 0.13 dyne/cm^2^, and it is within the range of shear stress that a normal kidney receives. Therefore, the renal function was evaluated by co-culturing HUVECs and RPTECs at an angle of 4° and an interval of 16 min, which is a condition of receiving appropriate shear stress for 14 days [14].

### 2.3. Cell Viability Assessment and Imaging

Before evaluating the secretory function of cells with chips, we observed cell growth images to confirm that cells in this model grow normally. Cells cultured with the insert for 14 days were observed with live and dead (L&D) staining. To dye live cell and dead cell together, the calcein AM (green, LIVE) and ethidiumhomodimer-1 (ethd-1) (red, DEAD) reagents were mixed in an appropriate proportion and dyed for 30 min. The confocal microscope was observed at 4× and 20× magnifications for imaging purposes. We analyzed the cell survival rate using Image J.

### 2.4. Immunofluorescence

RPTECs present in the proximal renal tubule have proteins that are essentially expressed. ZO-1 (zonulaoccludens-1) is a protein also known as tight junction protein-1 that connects transmembrane proteins to cytoskeletal proteins, actin. The protein is located on the cytoplasmic membrane surface at the hard intercellular junction. Sodium-dependent glucose cotransporter (SGLT2) is a protein responsible for the reabsorption of glucose and sodium in the kidney; SGLT2 suppression can reduce reabsorption and lower blood sugar levels.

The insert was separated from the chip and placed in 4% paraformaldehyde, and the cells were maintained at room temperature for 20 min. The cells were then permeabilized by adding 0.2 % (*v*/*v*) Triton-X100 (Sigma-Aldrich) in PBS and incubated for 20 min at room temperature. Subsequently, the cells were washed with a washing solution and incubated with a blocking solution (3% (*w*/*v*) bovine serum albumin (BSA; Sigma-Aldrich, Burlington, MA, USA)) for 40 min at room temperature. Primary antibodies were directed against ZO-1 (Invitrogen (Waltham, MA, USA), rabbit polyclonal, human, 61–7300) and SGLT2 (Abcam (Cambridge, UK), rabbit polyclonal, human, ab85626) overnight at 4 °C, which is followed by incubation with a fluorescently labeled secondary antibody (Alexa Fluor 488, Invitrogen, Goat anti-Rabbit IgG, A11034) for 2 h. After washing with PBS, the cells were stained with Hoechst (Thermo Scientific, Waltham, MA, USA) for 20 min at room temperature. The stained samples were observed under a fluorescence microscope (IX73, Olympus (Tokyo, Japan)) or confocal fluorescence microscope (K1-Fluo; Nanoscope Systems Co., Ltd. Taejeon, Korea).

### 2.5. Glucose and TEER

Glucose reabsorption was measured using a glucose meter (Accu-Chek Performa, Roche Diagnostics, Kwai Chung, Hong Kong) and test strip. In the glucose reabsorption assay, the glucose level of the medium was measured using a commercial glucose meter (Accu-Chek Performa; Roche Diagnostics). After analysis, 20 μL of the medium was collected in the medium chamber of the chip and dropped on a glucose meter test strip to measure the glucose concentration.

Trans-epithelial electrical resistance (TEER) is a well-known quantitative technique for measuring the integrity of tight junction dynamics in endothelial and epithelial fault cell culture models. Prior to evaluating the transport properties of drugs and chemicals, TEER levels are used as a reliable indicator of cell barrier integrity. TEER measurements have the advantage of being able to be performed in real time without cell damage. To measure TEER, an insert containing cells cultured on a chip for 14 days was transferred to a 6-well plate and then filled with badges. As TEER values can be affected by external factors such as temperature, paper discharge form, and cell algebra, the measurement was conducted with maximum error [22].

### 2.6. Permeability Measurement

The cells were removed from the culture medium of the culture chip and washed with Hank’s balanced salt solution (HBSS) for 30–60 min. The fluorescent solution used for permeability analysis (FITC-dextran; Sigma) was diluted with HBSS, and the prepared solution was placed in the chamber. HBSS that does not contain fluorescence was used in the insert. The insert fluorescence concentration that changed over time was measured. For the changing fluorescence concentration, a standard curve was created using a fluorescence spectrophotometer, and then, the concentration was determined using the measured intensity value. The wavelength was set in the range of 480–520 nm for the measurement. We evaluated how much fluorescence was transmitted through fluorescence, which changed every 6 h [23].

### 2.7. Evaluation of Metformin Secretion and Cimetidine Inhibition Using HPLC

High-performance liquid chromatography (HPLC) was performed to quantify metformin secretion. The excellent reproducibility of analytical values is exhibited by high-precision equipment capable of qualitatively and quantitatively measuring various analytic substances in the sample rapidly and simultaneously. It has the advantage of being able to measure multiple samples. After removing all the inserts and badges from the chamber that has been cultivated with chip for 14 days, 250 µL of metformin is treated in the chamber, and 300 μL of fresh badges is filled into the insert. After 6, 12, 24, and 48 h, the badges that come up on the insert are collected. Metformin concentration was determined by analyzing the badges with HPLC. In an experiment to determine the effect of cimetidine on a control group, the chip was treated with 0.1 mM cimetidine two hours in advance, and 100 µM metformin was processed to create an environment in which metformin could be inhibited.

### 2.8. qPCR (Quantitative RT-PCR) for Confirmation of Quantitative Transporter Expression

Transporters are present in the proximal tubule. The expression of organic cation transporter 2 (OCT2), multidrug and toxin extrusion 1 (MATE1), MATE2-K, and novel organic cation transporter type 1 (OCTN1) as representative transporters of the proximal renal ureter was evaluated. In order to quantify these transporters, cells were raked out, RNA was extracted, and cDNA synthesis was accelerated. Subsequently, RT-PCR was performed using AccuPower PCR PreMix (BIONEER, Daejeong, Korea), and RNA expression was confirmed through electrophoresis. We then proceeded with qPCR for quantitative analysis. The primer pairs for each gene target are listed in Table 1.

## 3. Result and Discussion

### 3.1. Comparison of Cell Cultures in Shear Stress and Membrane Pore Size

To attach cells to the membrane, the ECM was coated with collagen I. Before confirming whether the cells accurately represented the proximal renal tubule in insert-type chips, RPTEC and HUVEC were cultured in transwell (TW) and I-M chip for 14 days when the pore sizes were 0.4 μm and 3 μm, as depicted in Figure 2 and Figure 3. Green indicates LIVE cells in the L&D image. Overall, the cells were successfully attached to the membrane and cultivated well. Immunofluoresence observed SGLT2, which is responsible for transporting cell junction proteins and sodium-glucose transport proteins present in RPTEC.

### 3.2. Comparison of Pore Size with Static/Fluidic Cell Growth

Cells were cultured using I-M chip and TW with 0.4 μm and 3 μm membrane pore diameters. Cell viability, TEER, permeability and glucose reabsorption were measured and compared. All TWs and chips were seeded under the same conditions. Cell viability measurements calculated the ratio of living to dead cells through ImageJ using photos taken in 20× during L&D shooting. When comparing cell viability, TW and I-M chip were cultivated well, and there was no significant difference, as shown in Figure 4a. TEER values were evaluated by comparing the difference in electrical resistance inside and outside the insert with one of the chopstick machines using Millicell ERS-2 Voltohmeters (Milipore, MA, USA). The TEER values in the chip with a 3 μm membrane tended to slightly increase compared to the TW (Figure 4b). Therefore, to check the barrier integrity, it is better to use a higher membrane pore size (3 μm). To reduce the errors that may occur during TEER measurement, the cell resistance value was measured using the same volume and temperature of the medium used and the absence of cells as the control. For permeability, FITC-dextran fluorescent dye was diluted to 0.5 mg/mL and treated in chambers at both ends of the chip. After the 6 h treatment time had elapsed, the solution secreted above the membrane was collected, and the fluorescence was measured to confirm the transmittance. To check the permeated concentration, the permeability was calculated using a standard curve. As shown in Figure 4c, the chip demonstrated lower fluorescence transmittance compared to the TW. When cultured on the chip, the cells spread more evenly, and the fluorescence transmittance is low. In addition, when 0.4 μm and 3 μm were compared, it was confirmed that more fluorescence was transmitted in the chip with a larger pore size.

### 3.3. Confirmation of the Effects of Metformin and Cimetidine According to Pore Size

In this chip, it should be possible to delineate the proximal tubules and blood vessels where reabsorption and secretion take place, demonstrating the renal ADME process. The secretion of unfiltered substances from the blood vessels into the tubules occurs by the OCT2 transporter present in the basal of the tubules.

Therefore, as the role of the OCT2 transporter is important to reproduce the renal function, it is necessary to examine the transporter’s formation. To determine whether the secretion is carried out properly, the secretion ability of metformin and cimetidine was evaluated using HPLC. As shown in Figure 5, for the 0.4 μm membrane, the TW showed higher metformin concentration than the chip. Due to the large amount of metformin that permeated through the membrane, it was discovered that the cells on the chip were better cultured throughout the membrane. Conversely, cimetidine treatment increased the concentration in the TW, whereas the chip showed no significant difference. When metformin secretion at 3 μm was evaluated, the metformin concentration in the TW was comparable to that in the 0.4 μm membrane. Cimetidine showed an inhibition of metformin secretion when compared with 0.4 μm, but no significant inhibition was observed when compared to the effect of treatment with metformin alone under the same conditions.

### 3.4. Confirmation of Transporter Expression by qPCR

The secretory process of moving from a blood vessel to a tubule is carried out by OCT2 present in the basal of the tubule and MATE1, MATE2K, and OCTN1 transporters present in the apical end of the tubule [24,25,26]. To reproduce the secretory function, it is necessary to determine whether the expression of the transporter in charge of secretion by kidney cells is successful [27]. After confirming the presence of transporters, metformin was evaluated to ensure proper secretion. Metformin is a representative diabetes treatment agent secreted by blood vessels into the proximal tubule and acted upon by OCT2, which is a transporter present in the proximal tubule apical. Metformin secretion evaluation can be used to confirm the expression of OCT2 in 14-day-old cultured cells [28]. Then, we treated the chip with cimdetidine, an inhibitor of metformin, to determine whether secretion and inhibition can be reproduced [29].

The OCT2 transporter present in the basal of the tubule causes a secretion of unfiltered substances from the blood vessel to the tubule. In addition, MATE1, MATE2-K, and OCTN1 transporters were excreted in the urine from the proximal tubule [30]. Quantitative PCR (qPCR) was performed using Glyceraldehyde-3-phosphate Dehydrogenase (GAPDH) as a control to quantitatively confirm the amount of RNA present in the cell. As shown in Figure 6, the overall expression of the transporter was higher in the 3 μm membrane with a larger pore size, and more expression was shown in the I-M chip than in the TW. In particular, it was confirmed that the expression of MATE1 and MATE2K was significantly higher. In the case of OCT2, which acts on metformin secretion, the expression level was slightly higher in the I-M chip, but it was not significant, and the amount was lower than that of other transporters.

The PK test confirmed that metformin secretion was achieved using a metformin drug mediated by OCT2, and cimetidine, which inhibits metformin secretion, was used as a comparison for double confirmation. When the membrane pore size was large, it was determined that the level of metformin secretion was high and less than that of the TW. It was difficult to discern the effect of using the metformin inhibitor, cimetidine. However, the qPCR results can provide a comprehensive explanation for this issue. From qPCR, it was confirmed that the expression levels of MATE1, MATE2, and OCTN1 were significantly greater in the chip compared to TW. In contrast, the difference in expression level of OCT2 was not significant. It is presumed that it was difficult to observe the effect of cimetidine through the PK test because OCT2 was not properly formed. Therefore, in order to obtain a meaningful PK test result, it is believed that the effect of cimetidine can be confirmed with RPTEC cells amplified with OCT2 in a future experiment.

## 4. Conclusions

In this study, HUVECs and RPTECs were cultured for 14 days under four conditions based on the membrane pore size and the presence or absence of shear stress. To determine whether the cells grow normally on this chip, images of the cell junction protein ZO-1 and the glucose transporter SGLT2 were examined by Live & Dead staining and immunofluorescence. Under all conditions, cells were evenly distributed and grew on all surfaces of the membrane. Cell viability, TEER, glucose, and permeability, which can quantitatively confirm cell growth, were evaluated, and normal cell culture was observed on the chip. As a result, it was confirmed that the cell viability was higher in the 3 μm pore size than in the 0.4 μm, the cell culture performed better, and the drug secretion was enhanced when the pore size was large. The injection-molded polycarbonate microfluidic chip is anticipated to be commercially viable for drug-screening devices, particularly ADME tests.

## Figures and Tables

**Figure 1 micromachines-13-01932-f001:**
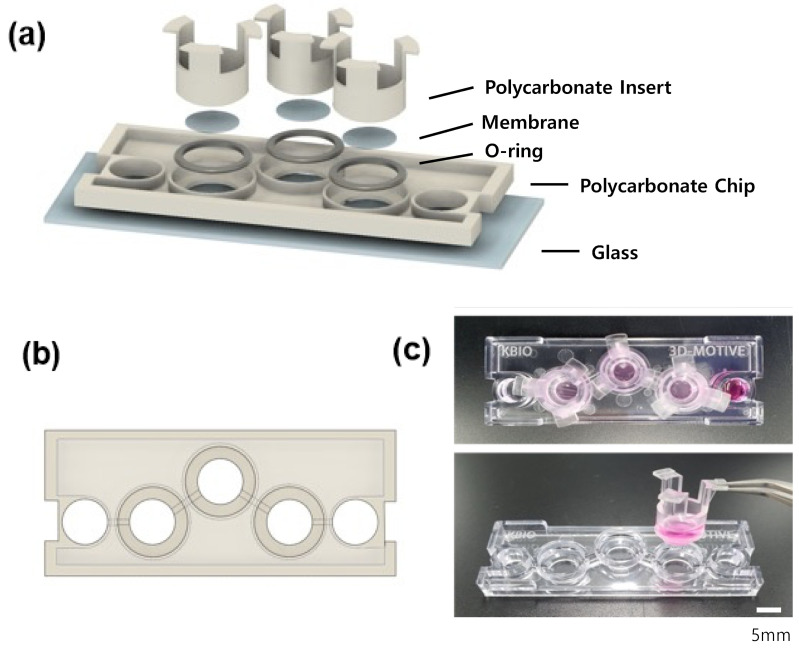
Injection-molded chip schematic and actual photograph. (**a**) Injection-molded chip with insert module installed. (**b**) Polycarbonate. (**c**) Actual photographs of injection-molded chip and insert.

**Figure 2 micromachines-13-01932-f002:**
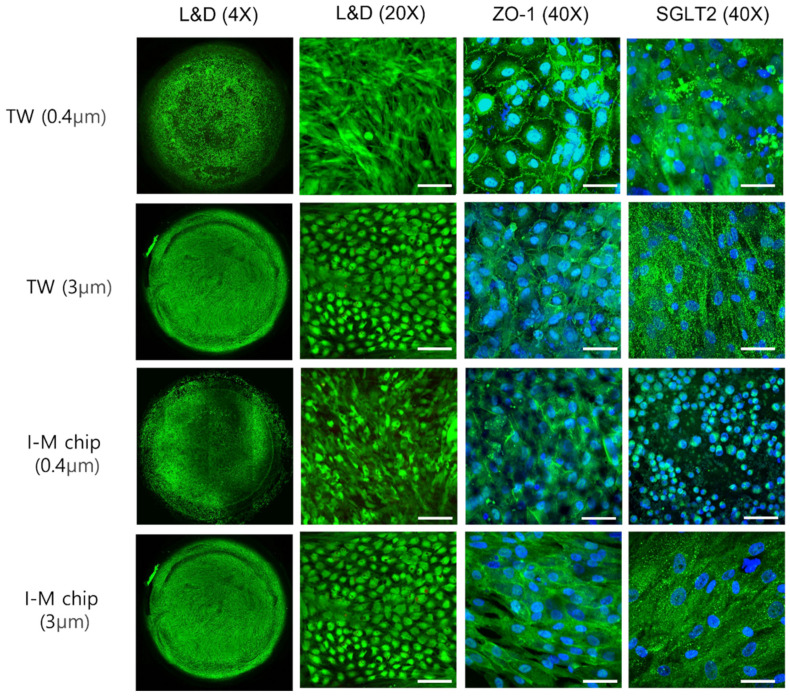
Using a confocal microscope, RPTEC’s L&D images (4× and 20×) and immunofluorescence images (40×) were observed in TW and I-M Chip using a confocal microscope, live–dead cell image observation throughout the membrane, L&D scale bar 100 µm. IF (ZO-1 and SGLT2) scale bar is 50 μm.

**Figure 3 micromachines-13-01932-f003:**
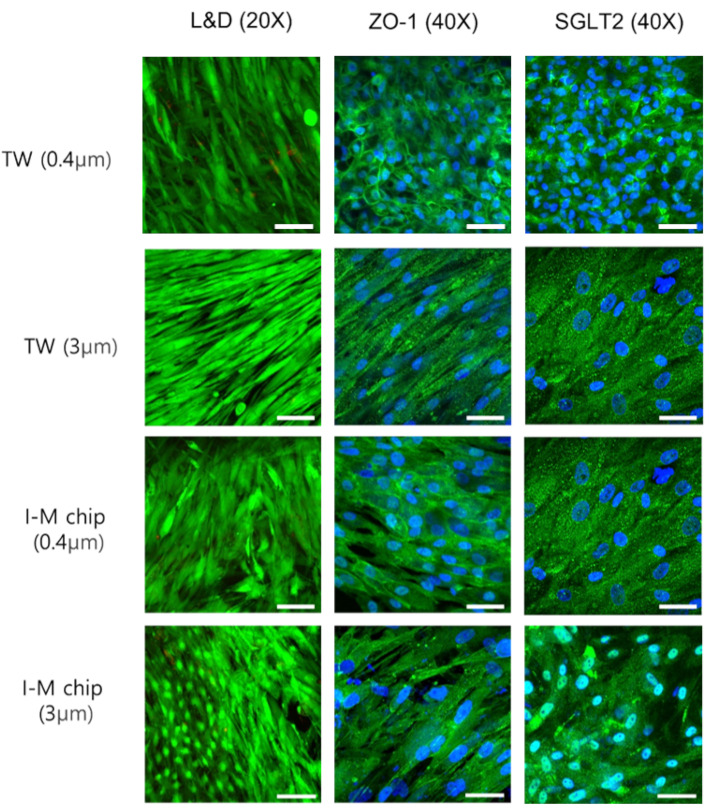
Confocal microscope HUVEC’s L&D images (4×, 20×) and immunofluorescence images (40×) on TW and I-M chip by membrane pore size.

**Figure 4 micromachines-13-01932-f004:**
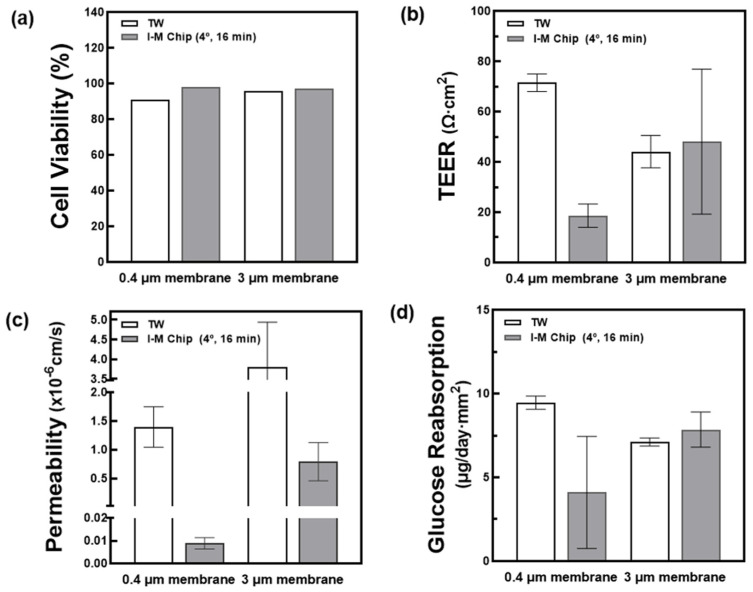
Confirmation of cell growth according to membrane size (0.4 μm, 3 μm). (**a**) Confirmation of cell viability through imageJ of images taken with L&D. (**b**) TEER measurement using a Millicell ERS-2 Voltohmmeter. (**c**) After 6 h of treatment with FITC–Dextran fluorescent dye at a concentration of 0.5 mg/mL, the transmittance was confirmed by measuring the absorbance (FL) of the fluorescence secreted above the membrane. (**d**) Glucose value of the medium 24 h after filling the TW and the chip with fresh medium was measured. Through this, the amount of glucose reabsorbed over 24 h was confirmed.

**Figure 5 micromachines-13-01932-f005:**
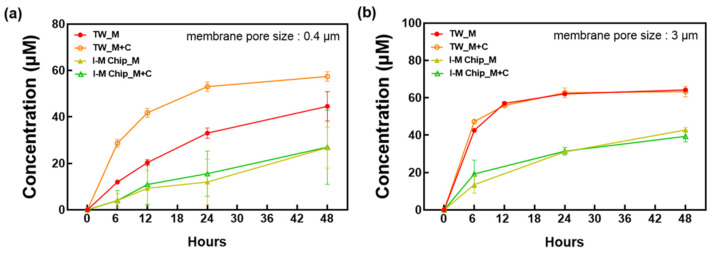
After culturing RPTEC and HUVEC on a TW and I-M chip for 14 days, metformin (100 μM) alone or metformin (100 μM) and cimetidine (0.5 mM) were treated with metformin secretion concentration after 6, 12, 24, and 48 h evaluated via HPLC. The difference between static TW and fluidic chip according to membrane pore size of (**a**) 0.4 μm and (**b**) 3 μm were compared. TW, C: culture method, R + H: RPTEC and HUVEC culture.

**Figure 6 micromachines-13-01932-f006:**
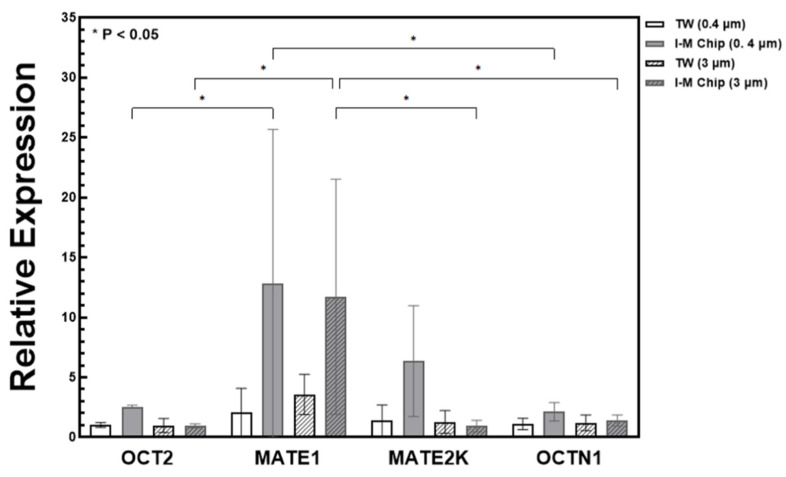
Quantitative PCR (qPCR) expression comparison results analysis. Relative expression levels of OCT2, MATE1, MATE2K, and OCTN1. Overall, it was shown that the gene expression level in chip was increased compared to transwell (* *p* < 0.05).

**Table 1 micromachines-13-01932-t001:** Primer sequence for each gene target.

Gene	Forward	Reverse
OCT2	GAGATAGTCTGCCTGGTCAATGC	GTAGACCAGGAATGGCGTGATG
MATE1	TGCTGTAGCCTTCAGTGTCCTG	GCTTCAAAGCGGTGGGAAACAGC
MATE2K	GCCTTTGGTGCCGCTGTGAATG	AGCAGTTGCCAGGAAGACACAG
OCTN1	TGGACCTGTTCAGGACTCGGAA	TAGGAGCATCCAGAGACAGAGC

## Data Availability

Not applicable.

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
