# Peer review of "Fabrication and Evaluation of Tubule-on-a-Chip with RPTEC/HUVEC Co-Culture Using Injection-Molded Polycarbonate Chips"

_micromachines, 2022, doi:10.3390/mi13111932_

Round 1
Reviewer 1 Report
The authors describe the fabrication of a polycarbonate microfluidic well chip for evaluation of a co-culture system of renal and endothelial cells and drug absorption specifically . The chip is a viable evaluation tool for kidney/blood transport on a chip.
The authors give a good introduction to the nephrological system for the laymen not familiar with kidney transport and waste clearance. The authors also gave a thorough background on current technologies in the field, including kidney on a chip. In this study, the authors are using an alternative polymer that would better suit manufacturability and biocompatibility concerns.
Overall, the manuscript only had a few minor issues that should be addressed prior to publication.
(1) Is this the identical device as described in ref 15? If so, you need to provide the reference? this is the first time with renal and endothelial cell (?) but the device fab is the same.
(2) On page 3, lines 129-131. The authors refer the readers to a previous paper that evaluated the shear stress and conditions for culture. The authors should give brief data to support the angle and time intervals. Also in the previous paper, there is no mention of an extension function - what do the authors mean by this term? This review has used this term in programming and strings, but its not clear what is meant here.
Also, what is the appropriate shear stress for physiological proximal tubules?
(3) The authors are making several references to static versus shear stress conditions. Are the I-M chips the only chips that experienced shear stess? Can we assume that TW are all static? Throughout the manuscript the authors make reference to shear stress (including the conclusion) but clearly is needed? Also pore size made a difference but was all four conditions compared static vs dynamic flow (basically 8 conditions; all four membranes and flow and no flow?). Please make clear throughout the manuscript where shear stress being applied.
Reviewer 2 Report
This manuscript presents an injection-molded polycarbonate microfluidic chip for co-culture of RPTEC and HUVEC cells to replicate in vivo kidney conditions. The microfluidic chip was fabricated by injection molding three badge chambers fit with O-ring inserts and integrated with membrane. RPTEC and HUVEC cells were cultured and seeded into membrane and later inserted into the microfluidic chip. Cell growth on-chip was monitored and assessed for the ratio of live to dead cells to gather their survival rate. The authors also use immunofluorescent methods to detect certain genes within the cell that indicate certain functions: absorption, secretion, metabolism, and distribution. They also tested for these functions by means of a glucose meter (glucose reabsorption), fluorescence for permeability of the cell, high-performance liquid chromatography to quantify secretion, and distribution by q-PCR test for certain transporter genes. The main advantage of this method is the usage of PC instead of PDMS, which seems preventing the adsorption of small molecule drugs. The authors states that their method can be anticipated to be commercially viable for drug screening devices, particularly ADME tests. The authors need to address the following comments before this manuscript can be considered for publication.
1, The main design purpose of the microfluidic chips and how it relates to the urinary system are not totally clear. How exactly is the chip related to the four kidney functions, namely distribution, metabolism, absorption, and excretion? It seems from the result this chip is mostly related to two of the functions, instead of all four. The authors could articulate more clearly of the necessity and the uniqueness of the chip they designed.
2, The manuscript could use some revision to be more reader-friendly. Sentences and paragraphs, especially in the introduction, could be tighten up with better logic links. A thorough revision of the language is recommended before the manuscript can be considered for publication.
3, More detailed descriptions of the cell and biological reagents used in their experiments are recommended. For example, what are the cell sources? What are the important growth factors used in this study? These information will help readers to repeat their experiments and help broaden the significance of this study.
Overall, I think this manuscript is interesting to fit the scope of Micromachines, but should be considered for publication only after revision.
Author Response
- The main design purpose of the microfluidic chips and how it relates to the urinary system are not totally clear. How exactly is the chip related to the four kidney functions, namely distribution, metabolism, absorption, and excretion? It seems from the result this chip is mostly related to two of the functions, instead of all four. The authors could articulate more clearly of the necessity and the uniqueness of the chip they designed.
Response :
Thank you for your helpful comment.
The ADME process is essential in the human body. However, not all of these processes occur in one organ, but multiple organs such as intestine, liver, and kidney can combine to realize ADME. The kidneys are the main organ responsible for the removal (reabsorption) of drugs and their metabolites. Therefore, in this study, reabsorption and secretion capacity were evaluated.
- The manuscript could use some revision to be more reader-friendly. Sentences and paragraphs, especially in the introduction, could be tighten up with better logic links. A thorough revision of the language is recommended before the manuscript can be considered for publication.
Response:
Thank you for your helpful comment. According to the reviewer’s comment, we tightened up the introduction and highlighted red color in the revised manuscript at line 43~55, 67~76.
- More detailed descriptions of the cell and biological reagents used in their experiments are recommended. For example, what are the cell sources? What are the important growth factors used in this study? These information will help readers to repeat their experiments and help broaden the significance of this study.
Response:
Thank you for your helpful comment.
Human RPTEC (renal primary tubule epithelial cell) (LONZA CC-2553) was used. Human Umbilical Vein Endothelial Cells (HUVEC) (LONZA C2519A) were used. We added it into the revised manuscript at line 95~97.